# Electrodeposition of Nanoparticles and Continuous Film of CdSe on n-Si (100)

**DOI:** 10.3390/nano9101504

**Published:** 2019-10-22

**Authors:** Walter Giurlani, Vincenzo Dell’Aquila, Martina Vizza, Nicola Calisi, Alessandro Lavacchi, Alessia Irrera, Maria Josè Lo Faro, Antonio Alessio Leonardi, Dario Morganti, Massimo Innocenti

**Affiliations:** 1Dipartimento di Chimica, Università degli Studi di Firenze, via della Lastruccia 3, 50019 Sesto Fiorentino, Italy; walter.giurlani@unifi.it (W.G.); vincenzo.dellaquila@stud.unifi.it (V.D.); martina.vizza@stud.unifi.it (M.V.); nicola.calisi@unifi.it (N.C.); 2CNR-ICCOM, Istituto di Chimica dei Composti OrganoMetallici, via Madonna del Piano 10, 50019 Sesto Fiorentino (FI), Italy; alessandro.lavacchi@iccom.cnr.it; 3CNR-IPCF, Istituto per i Processi Chimico-Fisici, V.le F. Stagno D’Alcontres 37, 98158 Messina, Italy; irrera@its.me.cnr.it (A.I.); antonio.leonardi@ct.infn.it (A.A.L.); dario.morganti@ct.infn.it (D.M.); 4Dipartimento di Fisica ed Astronomia, Università di Catania, Via Santa Sofia 64, 95123 Catania, Italy; mariajose.lofaro@ct.infn.it

**Keywords:** CdSe, cadmium, selenide, silicon, electrodeposition, emitting diode, optoelectronics, nanoparticles, thin film

## Abstract

CdSe electrodeposition on n-Si (100) substrate was investigated in sulfuric acid solution. The behaviour and the deposition of the precursors (Cd and Se) were studied separately at first. Then, we explored both the alternated deposition, one layer by one, as well as the simultaneous co-deposition of the two elements to form the CdSe semiconductor. Varying the deposition conditions, we were able to obtain nanoparticles, or a thin film, on the surface of the electrode. The samples were then characterised microscopically and spectroscopically with SEM, XRD and XPS. Finally, we evaluated the induced photoemission of the deposit for the application in optoelectronics.

## 1. Introduction

Silicon is both a very abundant and extremely interesting element from a technological point of view. Despite its diffusion in the microelectronic industry, the realization of Si-based devices is extremely challenging due to its indirect bandgap. In recent years, several strategies were proposed to overcome this issue, such as defect emission [1], fluorescent rare earth doping [2], and the quantum confinement effect [3]. In particular, silicon nanostructure, such as porous Si (p-Si), Si nanocrystals (Si NCs) and silicon nanowires (Si NWs) with suitable sizes for quantum confinement effect, emerged in recent years. However, a suitable candidate to integrate light in silicon and open the microelectronic world to the light remains to be found. Indeed, p-Si is commonly non-mechanically robust and characterised by a poor photoluminescence (PL) [4,5]. Si NCs were recently adopted for in vivo bioimaging, but an integration of these nanomaterials in a flat substrate compatible with Si industry without compromising their light emission property is very complex [6]. Moreover, the electrical pumping of such devices appears very challenging. In the recent years, the new fabrication method [7,8] opened the route to the realization of room temperature light-emitting Si NWs that were successfully employed in sensing [9,10,11,12], photovoltaics [13,14,15] and photonics [16,17].

However, in recent years, other strategies have emerged. The “More than Moore” philosophy has integrated new functionalities and characteristics to a single processor instead of just increasing its calculus power. Following the same idea, the integration of other semiconductors on silicon is gaining the attention of the scientific and industrial community. Indeed, coupling Si with other semiconductors is another interesting route to obtain light emission in Si-based devices and to integrate new functionalities that are typical of other materials [18].

Among semiconductors, other materials are emerging to face the important demand for the production of novel optoelectronic devices integrated in Si platforms. The possibility of integrating various materials on the Si surface offers the opportunity to produce compact and cost-effective electronic circuits with numerous applications [19,20,21]. Among the most common II-IV compounds, cadmium selenide (CdSe) is a semiconductor of particular interest in the optoelectronic field in the production of energy through solar cells [19] and in the use of light emitting diodes (LEDs) [21] and laser diodes [20]. Unfortunately, the mismatch of the lattice constants (11.4%) and the difference in thermal expansion make the CdSe epitaxial deposition on Si very complex with vapour phase techniques. Through the electrodeposition, it is possible to carry out a deposition with a fine control of the growth conditions by operating at low temperature and atmospheric pressure with a considerable financial gain. Furthermore, electrochemical processes can be scaled to industrial level and large areas with relative simplicity. Electrodeposition on Si is a not trivial task since the semiconductor nature of the substrate limits the electrochemical processes to a metal surface, greatly increasing the difficulty of a controlled deposition. The CdSe electrodeposition on various substrates is a well-explored topic [22,23,24,25,26,27,28,29,30], but there are not many attempts at growing this compound on silicon. Chubenko et al. succeeded in the deposition of CdSe from aqueous solution on porous silicon only at a high temperature and did not obtain satisfactory results on flat silicon [31].

In this work, we investigated the possibility of electrodepositing CdSe at room temperature on a commercial substrate of (100) n-doped silicon (Si-n (100)), both through the alternate deposition of a few atomic layers at a time of single precursors, employing the electrochemical atomic layer deposition (E-ALD) process [24,29,30,32], and through co-deposition. Initially, the behaviours of the Cd and Se precursors were studied separately on the substrate. Then, the deposition was carried out, optimizing the process with regard to different conditions. Finally, the samples obtained were characterized by microscopic and spectroscopic techniques.

## 2. Materials and Methods

### 2.1. Electrochemical Measurements

We used Na_2_SeO_3_ and 3CdSO_4_∙8H_2_O salts to prepare the Se(IV) and Cd^2+^ 0.1 mM solutions in H_2_SO_4_ 0.1 M using ultrapure MilliQ water (18 MΩ, Merk Millipore, Burlington, MA, USA). The solutions were deaerated with nitrogen and stored under nitrogen atmosphere in sealed Pyrex jars. In our automated deposition system [32], the jars were directly connected to the electrolytic cell, made of Kel-F, and the flow of the solution were controlled directly from the computer acting on electro-valves. The nitrogen pressure was regulated to obtain a flow rate of 1.5 mL/s. The capacity of the cell was 1.88 mL. The working electrode was a n-Si 100 (P-doped with a resistivity of 1–5 Ω∙cm) with a diameter of 1 cm. Before each deposition, the electrode was cleaned with the following procedure: Sonication in acetone (15 min) and subsequently in ethanol (15 min), immersion in HCl:H_2_O_2_:H_2_O solution (molar ratio 1:2:8) at 80 °C for 15 min, water wash, etching in HF solution (2 M) for 10 min, and rinsing in water. The electrochemical depositions were carried out at room temperature in the dark in order to exclude the influence of light that could potentially lead to the photoexcitation of silicon. All the given potentials refer to the Ag/AgCl sat. KCl electrode.

### 2.2. Microscopic and Spectroscopic Characterization

The scanning electron microscopy (SEM) images were acquired using a Gaia 3manufactured by Tescan (Brno, Czech Republic) with a sensor for secondary electrons. The energy was equal to 20 kV for image acquisition.

The composition of the CdSe films were evaluated by X-ray photoelectron spectroscopy (XPS). The ultra-high vacuum chamber of the instrument was equipped with a non-monochromatic X-ray source (VSW Scientific Instrument Limited model TA10 (Manchester, UK), Al Kα radiation, 1486.6 eV) and a hemispherical analyser. The source was set to work at 144 W (12 kV and 12 mA). The analyser makes use of a 16-channel detector and dedicated pumping system that permit the analyser to work with a relative high pressure in the main chamber of the instrument during the measurement (up to 10-7 mbar). For this analysis, the pass energy was set to 44 eV. The obtained spectra were analysed using the dedicate software CasaXPS (version 2.3.19, Casa Software Ltd, Teignmouth, UK). The background was evaluated and subtracted using the Shirley’s method [33]. For the fitting of the component, a mixed Gaussian-Lorentzian function was used. Spectra were calibrated, shifting the aliphatic component of carbon 1 s transition to 284.8 eV [34].

The crystallinity of the deposit was characterized using a Bruker (Billerica, MA, USA) New D8 Da Vinci Diffractometer to perform X-ray Diffraction spectroscopy (XRD) with Cu K radiation, Ni filter, fast multichannel energy-discriminator detector, flat holder and Bragg-Brentano configuration.

The room temperature emission of the samples was tested by photoluminescence spectroscopy using a Jobin Ivon Horiba Spectrometer equipped with a solid-state laser at the wavelength of 561 nm, which was focused onto the sample through a 100X (0.9 NA) objective. The room temperature emission of the samples was then analysed by a Peltier cooled CCD detector (Synapse). This setup worked in the backscattering configuration and the same objective was used to acquire the signal.

## 3. Results

The samples synthesized in this work are summarized in Table 1.

### 3.1. Cadmium

In order to understand the electrochemical phenomena underlying the CdSe electrodeposition, a study of the electrochemical behaviour of the precursors was necessary. The cyclic voltammetry (CV) (Figure 1a) of the cadmium on n-Si was investigated first: The reduction of the metal starts at −0.8 V and the reduction peak is visible at −0.83 V, while the oxidation peak is at −0.70 V. The anodic peak was much more intense than the cathodic one since, in the initial stages of the anodic scan, the reduction kept taking place. In fact, there was a cross between the two scans. This behaviour suggests an inertia in the deposition of Cd on Si: When the process started, the first nuclei were formed. This potential is known as nucleation overpotential [35,36,37,38,39]. During the anodic scan, the Cd^2+^/Cd couple was present and the oxidation took place near the Nernst potential (more positive), producing the overlap between the cathodic and anodic scan. From this consideration, we can assume that the deposition more likely led to separate clusters or nanoparticles (NP) than to a continuous thin film. We also investigated the possibility of a surface-limited reaction by performing depositions at various potentials and at a fixed time (60 s). From the stripping peak (Figure 1b inset), we extrapolated the deposited charge density (Figure 1b) by means of Equation (1) [40,41,42], where *ρ* is the charge density, *I* the current, *E* the potential, *v* the scan rate and *A* the electrode area. We did not observe any plateau in the charge density plot, suggesting that no surface-limited reactions (SLRs) were present for Cd on Si. The absence of the under potential deposition (UPD) peak in the CV advised the same conclusion.

(1)$$\rho =\frac{\int^{\;}IdE}{v\cdot A}$$

The electrodeposition of Cd (sample A) was performed with a charge-controlled deposition: −0.8 V was applied to the electrode and every 200 µC fresh Cd^2+^ solution was fluxed in the cell, and the process was repeated for 30 cycles (ca 30 nmol in total). The charge value was chosen close to the theoretical amount of a single monolayer of Cd on Si (100).

SEM analysis was performed on sample A (Figure 2a and Appendix A). As predicted, the deposit resulted in well-separated faceted islands made of two different family of crystals. The mean diameter of the first family was 0.17 ± 0.11 µm and the second was 0.93 ± 0.16 µm. The particles distribution is reported in Figure 2b with a low surface coverage of ≈3%, measured as the ratio of the cumulative area filled with island per the total scanned area.

### 3.2. Selenium

A different response was observed for the selenium deposition (sample B). The behaviour of selenium on n-Si is reported in Figure 3, where the reduction of Se(IV) began at −0.57 V.

In this case, no anodic peaks were detected. This was probably due to the high barrier between Se and n-Si [43], which limited the oxidation of the chalcogen.

We prepared the sample B by applying −0.8 V to the electrode. Every 400 µC fresh Se(IV) solution was fluxed in the cell. The process was repeated for 30 cycles (ca 30 nmol in total).

SEM analysis was performed on sample B (Figure 4a and Appendix A). In this case, the deposit was composed of a large spherical particle distribution with a mean diameter of 0.70 ± 0.50 µm. The measured particles distribution is reported in Figure 4b and a low surface coverage (≈8%) was calculated as described for sample A.

### 3.3. Cadmium Selenide

After the electrochemical characterization of Cd and Se separately, we performed an alternating deposition (sample C). It involved the deposition of cadmium in the cadmium solution. The cell was then washed with sulfuric acid, and selenium was deposited in selenium solution. The full procedure was performed automatically with our deposition system, which was connected to a computer without removing the electrodes from the cell. We investigated the electrochemical behaviour of Cd^2+^ on Se/n-Si (Appendix A) and Se(IV) on Cd/n-Si, and found a shift of the reduction potential to more positive values for both the element. We did not observe any anodic peaks in the case of Cd^2+^, suggesting the formation of a stable compound. Then, we performed the alternating deposition, in which a single cycle consisted in four steps: 200 µC deposition of Cd at −0.8 V; wash in H_2_SO_4_ solution; 400 µC deposition of Se at −0.8 V; wash in H_2_SO_4_ solution. The process was repeated for 30 cycles (ca 30 nmol in total).

SEM analysis of sample C (Figure 5a) revealed smaller particles and a higher coverage of the n-Si surface respect to sample A and B, which still left room for improvements. The dimension of the particles decreased considerably, with the mean of the distribution at 26 ± 18 nm. Even if the distribution became larger (Figure 5b), the coverage (≈9%) still was not enough for appreciable photoluminescence effects.

The possibility of the co-deposition process was investigated. We prepared a solution with both Cd^2+^ 0.1 mM and Se(IV) 0.1 mM in H_2_SO_4_ 0.1 M. In the resulting cyclic voltammetry on n-Si (Figure 6), the reduction processes started at −0.5 V. Then, two cathodic peaks were present: The first at −0.81 V assigned to the formation of CdSe, and the second at −0.91 V for the reduction of bulk Cd. During the anodic scan, no oxidation peak was detected, suggesting an irreversible process. Even in this case, we observed a cross in the voltammogram due to the overpotential in the deposition on silicon.

We prepared two samples, D and E, applying −0.84 V (the potential between the two reduction peaks) to the n-Si electrode. Every 600 µC fresh co-deposition solution was fluxed in the cell, and the process was repeated 10 times for sample D (ca 10 nmol in total) and 30 for sample E (ca 30 nmol in total).

SEM analysis was performed on sample D (Figure 7a) and E (Figure 7b,c). In sample D, we obtained smaller NPs with a mean diameter of 61 ± 33 nm (reported in Figure 7b) and high surface coverage (≈65%). In sample E, the NP coalesced, forming a complete thin film all over the n-Si electrode.

### 3.4. Spectroscopic Characterizations

On all the samples, we performed an XPS analysis to evaluate the composition of the deposits. Sample A and B were used as references. The 3d transition of Cd (Figure 8a) in sample A was detected at 406.0 eV and attributed to Cd^2+^ [44,45], which was formed naturally on the Cd deposit when exposed to atmospheric oxygen. In sample C, D and E, the same transition shifts to 404.9, 405.1 and 405.3 eV, respectively, indicating the formation of the CdSe compound [46]. We found a further confirmation of the CdSe formation in the Se 3d region (Figure 8b): In sample B, we found the transition at 55.2 eV, compatible with Se(0) [47], while in sample C, D and E, the transition was detected at 54.0, 54.1 and 54.3 eV, respectively, compatible with Se(-II) [46].

We performed also XRD (Figure 9a) and PL (Figure 9b) characterization on sample E. Since the amount of material was too low to obtain an appreciable signal, samples C and D were not suitable for these measurements. From the XRD analysis, we concluded that the deposit had a sphalerite (cubic zincblende) structure. Indeed, the three typical peaks (111, 220 and 311) of this form were visible. The sphalerite form was not the most stable crystalline structure for CdSe and converted to wurtzite over 130 °C. However, this structure is the typical form obtained from low-temperature aqueous synthesis [48,49]. The PL analysis highlighted an emission at 720 nm due to the presence of CdSe [50].

## 4. Conclusions

In this work, we successfully demonstrated the possibility to obtain CdSe NPs and thin films on n-Si commercial wafer at room temperature by a low-cost technique. We investigated both the alternate deposition of controlled amount of Cd and Se, as well as the co-deposition of the two elements. The second approach gave us the best results, providing a better coverage of the electrode surface and smaller particles. Modulating the number of the deposition cycles, it was possible to obtain both NP and a continuous film on n-Si. The formation of the compound was confirmed by XPS analysis. The deposit obtained was crystalline with a zincblende structure. The sample with continuous film of CdSe exhibited interesting PL emission, peaking at 720 nm at room temperature. This electrochemical approach opens the route toward more compact and cost-effective optoelectronic circuit production, with numerous applications in photonics and photovoltaics integrating properties of different semiconductors materials compatible with Si technology.

## Figures and Tables

**Figure 1 nanomaterials-09-01504-f001:**
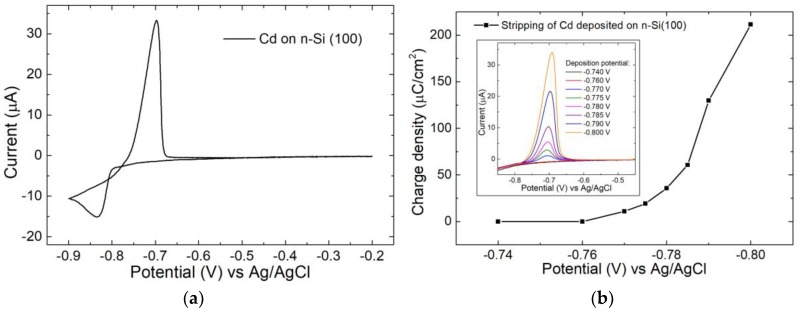
(**a**) Cyclic voltammetry (CV) of Cd^2+^ solution on n-Si in sulfuric acid between −0.2 V to −0.9 V, scan rate 10 mV/s; (**b**) charge density of Cd deposited varying the deposition potential, calculated from the stripping voltammetry (inset) performed between −0.85 V and −0.4 V, scan rate 10 mV/s.

**Figure 2 nanomaterials-09-01504-f002:**
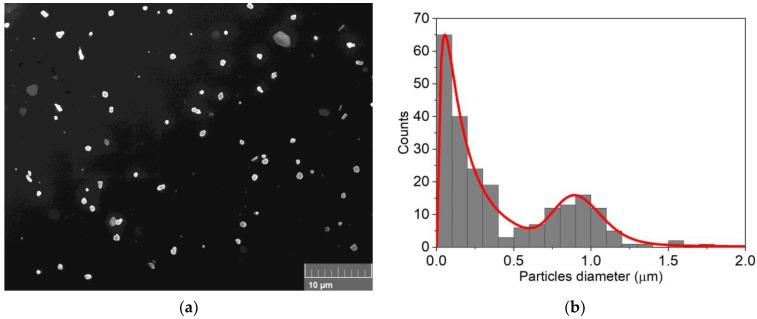
(**a**) SEM analysis of 30 nmol of Cd deposited on n-Si (sample A); (**b**) particles distribution obtained from the SEM image.

**Figure 3 nanomaterials-09-01504-f003:**
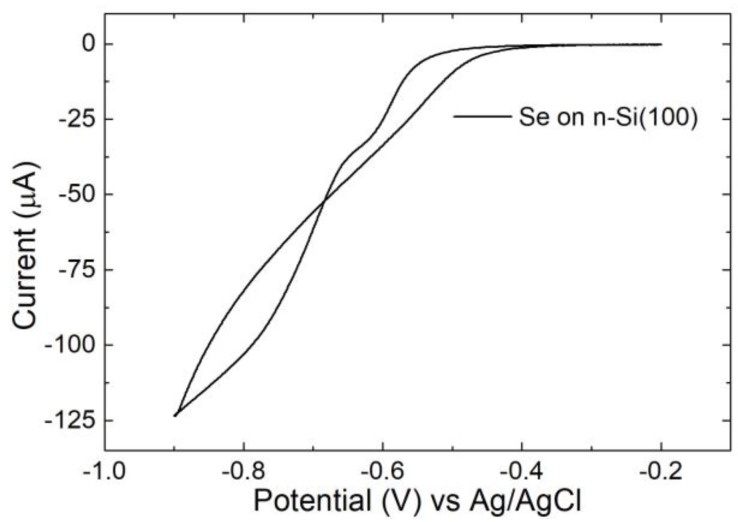
CV of Se(IV) solution on n-Si in sulfuric acid between −0.2 V to −0.9 V, scan rate 10 mV/s.

**Figure 4 nanomaterials-09-01504-f004:**
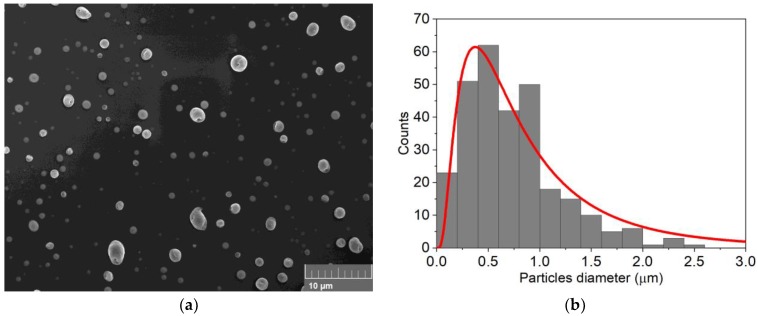
(**a**) SEM analysis of 30 nmol of Se deposited on n-Si (sample B); (**b**) particles distribution obtained from the SEM image.

**Figure 5 nanomaterials-09-01504-f005:**
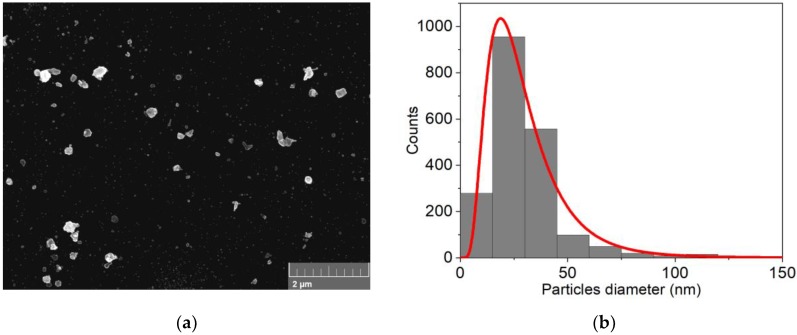
(**a**) SEM analysis of 30 nmol of CdSe deposited on n-Si with the alternating method (sample C); (**b**) particles distribution obtained from the SEM image.

**Figure 6 nanomaterials-09-01504-f006:**
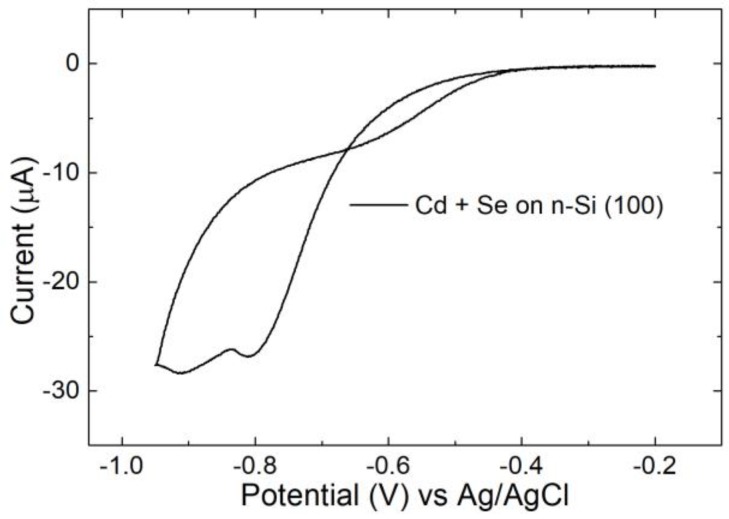
CV of Cd^2+^ and Se(IV) solution on n-Si in sulfuric acid between −0.2 V to −0.95 V, scan rate 10 mV/s.

**Figure 7 nanomaterials-09-01504-f007:**
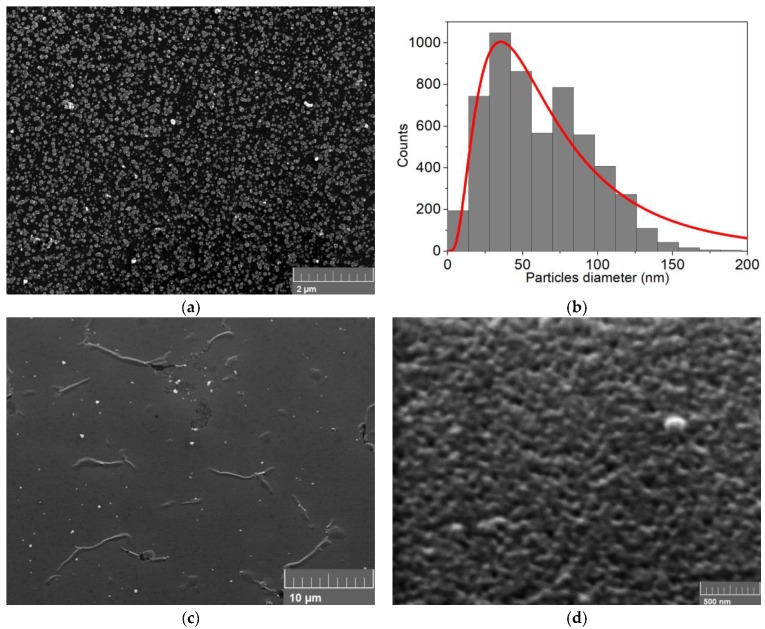
(**a**) SEM analysis of 10 nmol of CdSe co-deposited on n-Si (sample D); (**b**) particles distribution obtained from the SEM image a; (**c**) SEM analysis of 30 nmol of CdSe co-deposited on n-Si (sample E); (**d**) particles distribution obtained from the SEM image c.

**Figure 8 nanomaterials-09-01504-f008:**
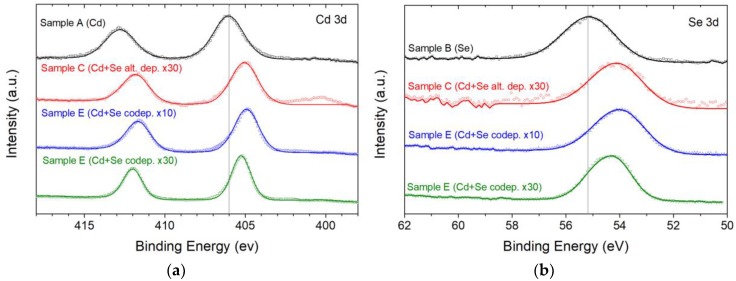
XPS analysis of the five samples: (**a**) High-resolution spectra of the Cd 3d zone (sample A, C, D and E); (**b**) high-resolution spectra of the Se 3d zone (sample B, C, D and E).

**Figure 9 nanomaterials-09-01504-f009:**
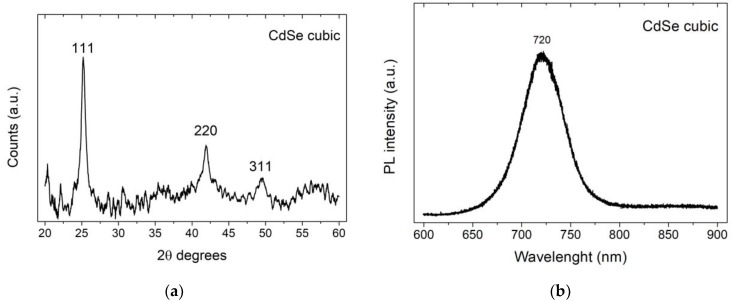
Spectroscopic analysis of sample E: (**a**) XRD analysis, coincides with the typical pattern of the zincblende form; (**b**) PL spectra, excited with a 561 nm laser, with the characteristic peak at 720 nm.

**Table 1 nanomaterials-09-01504-t001:** Scheme of the samples synthesized in this work.

Sample	Description
A	Cd NPs, 30 nmol deposition
B	Se NPs, 30 nmol with deposition
C	CdSe NPs, 30 nmol with alternate deposition
D	CdSe NPs, 10 nmol with co-deposition
E	CdSe thin film, 30 nmol with co-deposition

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
