# Peer review of "Electrodeposition of Nanoparticles and Continuous Film of CdSe on n-Si (100)"

_nanomaterials, 2019, doi:10.3390/nano9101504_

Round 1

Reviewer 1 Report

Nanomaterials 615978

Title: Electrodeposition of nanoparticles and continuous 2 film of CdSe on n-Si (100)

The authors present the obtaining of CdSe nanoparticles and thin films on polished Si commercial wafer. The state of the art regarding the CdSe thin films is very well addressed.

The manuscript is generally well written, but there are some minor typing mistakes (line 218 "it is was")

Comments:

1: Can the authors show SEM images with higher magnification for fig 2.a and 4.a?

2: Can the authors introduce the CV for alternating electrodeposition of Cd and Se?

3: Please check the measure unit in fig 7.b.

In my opinion, this is a useful paper to be published.

Author Response

We thank the reviewer for the nice comments and suggestions.

We corrected the typing mistakes.

1: The SEM images of the samples reported in figures 2a and 4a were took also at higher magnification, we didn’t insert in the manuscript because only few particles fit in the image, not allowing to have a good statistic on the distribution. We add the two images in the supporting information Figure S1 and S2.

2: We made some clarification in the text regarding the alternating deposition. We also add the cyclic voltammetry of Cd solution on Se and of Se solution on Cd in the supporting information.

3: the particle size should be in nm, we corrected it, thank you.

Reviewer 2 Report

Electrodeposition of CdSe has been quite broadly studied, in this paper the direct electrodeposition onto Si, and a study of the conditions, is a useful exercise of broad interest to readers of this journal, and is thus of sufficient interest to publish

I would like to see the authors address a few issues before publication:

-Solution C gave CdSe nps with alternate deposition, XPS indicated CdSe and not separated Cd and Se, but XPS is only a surface characterisation, how do the authors know that homogeneous nps of CdSe have been formed?.

Fresh Cd solution (line 133),  fresh Se solution (line 152), should be more specific, especially for Se, where SeO3 is present in solution, to remind readers of the nature of the electrodeposition precursors (i.e. free  Se ions are not present). For example the detail in line 173/4 is appropriate

Line 118: “There is a cross between two scans:”….”first nuceli are formed, then growth takes place”. This can be explained a little better (a couple of sentences), with reference to literature on nucleation and crossover on voltammograms where deposition takes place. It is seen throughout the paper, and so must be expanded and referenced properly.

Author Response

We thank the reviewer for the nice comments and suggestions, in particular we made these corrections:

1) If the deposition is limited to few atomic layer and if the formation energy of the compound is negative enough, the two element react to form the compound involving the E-ALD process. Other authors confirmed that in the case of Se and Cd this process take place. We have add some references in the introduction. We confirmed this behaviour performing CVs of Cd2+ on Se/Si and of Se(IV) on Se/Si (added to the supporting information) in which we didn’t observe any anodic peak for Cd, that we had observed in absence of Se, suggesting the formation of the compound.

 2) We substitute “Cd” and “Se” with “Cd2+” and “Se(IV)” when we talk of the ions in solution.

3) We tried to clarify the cross behaviour introducing the nucleation overpotential. We added also some references.